# ZBP1-Mediated Necroptosis: Mechanisms and Therapeutic Implications

**DOI:** 10.3390/molecules28010052

**Published:** 2022-12-21

**Authors:** Xin-yu Chen, Ying-hong Dai, Xin-xing Wan, Xi-min Hu, Wen-juan Zhao, Xiao-xia Ban, Hao Wan, Kun Huang, Qi Zhang, Kun Xiong

**Affiliations:** 1Department of Human Anatomy and Neurobiology, School of Basic Medical Science, Central South University, Changsha 430013, China; 2Xiangya School of Medicine, Central South University, Changsha 430013, China; 3Department of Endocrinology, Third Xiangya Hospital, Central South University, Changsha 430013, China; 4Department of Dermatology, Xiangya Hospital, Central South University, Changsha 430013, China; 5Ministry of Education, Key Laboratory of Emergency and Trauma, College of Emergency and Trauma, Hainan Medical University, Haikou 571199, China; 6Hunan Key Laboratory of Ophthalmology, Changsha 430013, China

**Keywords:** ZBP1, PANoptosis, pyroptosis, apoptosis, necroptosis

## Abstract

Cell death is a fundamental pathophysiological process in human disease. The discovery of necroptosis, a form of regulated necrosis that is induced by the activation of death receptors and formation of necrosome, represents a major breakthrough in the field of cell death in the past decade. Z-DNA-binding protein (ZBP1) is an interferon (IFN)-inducing protein, initially reported as a double-stranded DNA (dsDNA) sensor, which induces an innate inflammatory response. Recently, ZBP1 was identified as an important sensor of necroptosis during virus infection. It connects viral nucleic acid and receptor-interacting protein kinase 3 (RIPK3) via two domains and induces the formation of a necrosome. Recent studies have also reported that ZBP1 induces necroptosis in non-viral infections and mediates necrotic signal transduction by a unique mechanism. This review highlights the discovery of ZBP1 and its novel findings in necroptosis and provides an insight into its critical role in the crosstalk between different types of cell death, which may represent a new therapeutic option.

## 1. Introduction

Cell death is a fundamental pathophysiological process in various diseases. According to the type of death process, cell death can be divided into two major groups: programmed cell death (PCD), a precise and genetically controlled cellular death process, and non-PCD, also called necrosis. In past decades, PCD has been indicated to play important roles in the development of human diseases and immune response [1].

Apoptosis is the first programmed cell death pathway to be identified [2,3]. This cell death mainly occurs in the process of development and aging, while it can occur under a variety of pathological stimuli in the immune defense [4]. When apoptosis occurs, it shows cell shrinkage, condensation of the chromatin, formation of an apoptosome, and phagocytosis [5]. The execution of this pathway is considered to be related to the Bcl-2 protein family and the Cysteinyl aspartic acid protease (Caspase) family [6,7].

Necrosis, as opposed to apoptosis, refers to a passive death when cells are injured, which is characterized by cytoplasmic swelling, membrane rupture, and the release of intracellular contents [8]. Necroptosis is a form of regulated necrosis controlled by receptor-interacting protein (RIP) kinases (RIPKs) [9]. However, it is found that the tumor necrosis factor (TNF) pathway, which induces apoptosis, can also mediate the occurrence of necroptosis under certain conditions [10]. In addition, other PCD pathways can also occur along with necroptosis [11].

Pyroptosis is a new type of PCD found in recent years, which is a type of typical inflammatory cell death. It mostly occurs in infectious diseases [12]. Morphologically, the formation of membrane pores, breaking of the plasma membrane, and release of cell content cause a strong inflammatory response in pyroptosis [13]. Inflammasomes play a major role in the process of pyroptosis, which activates Caspase family members to promote the activation of pro-inflammatory cytokines IL and gasfermin protein. In recent years, it has been found that there is crosstalk between various PCD pathways, and the discovery of key factors that can widely regulate these processes is a research hotspot.

ZBP1, namely Z-DNA binding protein 1, was originally called DLM-1, which is the name of the gene it originally identified. It is a kind of tumor-related protein strongly induced by LFN-γ or lipopolysaccharides (LPS), and the study suggested that ZBP1 plays a role in host response in neoplasia [14]. Subsequent studies reported that the N-terminal of DLM-1 contains the same Z-DNA-binding domain (ZBD) as the RNA-editing enzyme adenosine deaminase acting on RNA1 (ADAR1), suggesting that DLM-1 may act as an intracellular DNA sensor [15]. The expression of ZBP1 is strongly induced by other IFNs, and selectively enhances the expression of DNA-mediated type I IFN and other innate immune-related genes [16]. Accordingly, it was designated as a DNA-dependent activator of IFN regulatory factors (DAI), suggesting that ZBP1 plays a significant role in DNA-mediated activation of the innate immune response. It connects pathogen-associated molecular patterns (PAMPs) and damage-associated molecular patterns (DAMPs) with intracellular pro-inflammatory signal transduction [17]. In terms of necroptosis, early studies focused on viral infection, which demonstrated that ZBP1, as a receptor of viral RNA (vRNA), triggered cell death pathways predominantly via necroptosis and inflammatory response [18]. In addition, the important functions of ZBP1 have also been confirmed in human diseases, including SARS-CoV-2 infection [19], cancer [20], and skin inflammation [21].

## 2. ZBP1, the Innate Sensor

### 2.1. Structure of ZBP1

ZBP1 contains two N-terminal ZBDs (Zα1 and Zα2), at least two RIP homotypic interaction motif domains (RHIM1 and RHIM2), and one C-terminal signal domain (SD) (Figure 1) [22]. Zα2 domain plays a key role in sensing Z-DNA and Z-RNA. Relevant studies demonstrated that specific mutations in this region effectively block the recognition of ZBP1 with vRNA or endogenous Z-NA, thereby inhibiting subsequent cell death and inflammation [23]. This domain is also the target of many ZBP1 inhibitors, including vaccinia virus (VACV) E3 protein and ADAR1 [24,25]. The RHIM domain mediates cell death. ZBP1 combines with receiver interacting protein kinase 3 (RIPK3) via the RHIM domain [26]. ZBP1 promotes RIPK3 autophosphorylation and induces phosphorylation of mixed linear kinase domain-like (MLKL), the downstream necroptosis executor, to induce necroptosis. In the presence of RIPK1, a protein with the same RHIM domain, the binding of ZBP1 to RIPK3 is inhibited by RIPK1 competition [27]. Murine cytomegalovirus (MCMV) M45 protein, which is a co-purified protein in virus and host immune defense system, also carries an N-terminal RHIM domain. It inhibits necroptosis by simulating the interaction between RIPK1 and RIPK3 to form a heterogeneous amyloid structure [28]. The SD domain of ZBP1 recruits TANK-binding kinase-1 (TBK1) and IFN regulatory factor 3 (IRF3) to activate type I IFN synthesis and other inflammatory reactions [29]. However, the ZBP1-IRF3 axis also mediates the proliferation of myeloma cells [30].

As a sensor of Z-NA, ZBP1 mainly relies on its Zα domain to identify ligands. In the middle part of ZBP1, there are at least two RHIM domains, which can bind with other RHIM-containing proteins (such as RIPK1, RIPK3, and TRIF) and mediate downstream signal transduction. These two special domains may also become targets for ZBP1 inhibition. For example, the M45 protein of MCMV can inhibit ZBP1-mediated cell death with its RHIM domain. While ADAR1-P150 is an inhibitor with ZBP1 by the Zα domain hindering the activation of ZBP1, it has a unique extra Zα domain, compared with the invalid subtype ADAR1-P110. Zα1, Zα2, Z-α, and Z-β are Z-DNA binding domains. SD: Signal domain; KD: Kinase domain; ID: Intermediate domain; DD: Death domain; TIR: Toll/interleukin-1 receptor domain; RNR-LIKE: Ribonucleotide reductase-like domain.

### 2.2. ZBP1 Binds Viral Z-NA to Mediate Inflammatory Response and Host Defense Response

The molecule most closely relevant to ZBP1 is undoubtedly IFN. ZBP1 expression is induced by IFN and also induces IFN responses [31]. This association with IFN suggests that ZBP1 plays an indispensable role in the inflammatory response and host defense [32]. Since ZBP1 contains ZBD, studies investigated the type of Z-DNA it binds to and the induced immune response [33]. Preliminary studies reported that both B-DNA and Z-DNA derived from multiple sources (synthetic DNA or DNA of bacterial, viral, or mammalian origin) induce strong expression of ZBP1 and IRF to mediate IFN expression and antiviral response [34]. The recognition of Z-RNA by ZBP1 of influenza virus (IAV) resulted in necroptosis [35]. Here, ZBP1 acted as an innate sensor of IAV recognizing Z-RNA in the viral ribonucleoprotein (vRNP) complex to induce necroptosis to resist virus infection. ZBP1 also induced interleukin-1α (IL-1α) in IAV via NOD-like receptor (NLR) family pyrin domain-containing 3(NLRP3) and recruited pulmonary neutrophils, resulting in inflammation [36]. Further studies proved that defective viral genes (DVGs) of IAV and other orthomyxoviruses produced Z-DNA, which were sensed by ZBP1, and induced cell death and inflammatory responses [37]. In addition, ZBP1 sense endogenous Z-NA in mice to induce cell death and skin inflammation, especially in the case of RIPK1 and Caspase-8 mutations [38]. ZBP1 acts as a cytoplasmic DNA receptor in many types of pathogenic infections, including *Toxoplasma gondii* infection [39,40], Fungi [41], and *Yersinia pseudotuberculosis* [42]. However, it remains to be confirmed whether Z-NA can be produced in these pathogens and other viruses for ZBP1 sensing.

### 2.3. ZBP1 Senses Endogenous Z-NA and Induces Cell Death

For a long time, studies have focused on the role of ZBP1 in sensing viral nucleic acid in virus-induced cell death, but whether ZBP1-mediated cell death in non-viral infections can detect endogenous ligands remains to be explored [43]. Jonathan et al. reported the recognition of endogenous nucleic acids in noninfectious cells with high expression of ZBP1 [44]. Further, photoactivatable ribonucleoside-enhanced crosslinking and immunoprecipitation (PAR-CLIP) analysis demonstrated that ZBP1 binds to RNA rather than DNA, and these nucleic acids may be in the Z-conformation. In this study, ZBP1 was affected by Caspase-8 to induce cell death, which may be mediated via RIPK3, which was obviously different from viral infection.

New progress was made in 2020 [38]. The team found that ZBP1 recognition of endogenous Z-NA triggered inflammation and cell death in RIPK1-deficient mice, which led to skin inflammation. In addition, ZBP1 can also sense endogenous ligands to trigger cell death resulting in colitis in mice by inhibiting FADD-Caspae-8 signal transduction [45]. ZBP1 binds to endogenous dsRNA via the Zα domain, which is most likely mediated by endogenous retroelements (ERE). In EREs, B2 and Alu SINEs have the greatest potential to form dsRNA [46]. They were specifically expressed in the epidermis and formed dsRNA to induce cell death and skin inflammation in RIPK1-deficient mice [21]. ADAR1 carries a Zα domain, which can edit dsRNA produced by ERE, suggesting that ADAR1 may play an important role in mediating the recognition of endogenous nucleic acid by ZBP1. In recent years, some studies reported the regulatory effect of ADAR1 on ZBP1-mediated cell death and inflammation and identified ADAR1 as a negative regulator of ZBP1 [47]. 

ADAR1 can be classified into two subtypes, P110 and P150. P150 can be induced by IFN and plays a major role in regulating ZBP1(Figure 2) [48]. Compared with P110, P150 contains additional Zα domains and nuclear output signals (NESs), which determine its ability to translocate into the cytoplasm and interact with ZBP1. The negative regulation of ADAR1 on ZBP1 occurs via inhibition of Z-RNA- and ZBP1-dependent cell death by preventing the accumulation of mRNA transcripts, which form Z-RNA [49]. However, it is directly associated with ZBP1 Zα domain interactions, which hinder the recognition of endogenous Z-NA. In ADAR1-deficient mice, ZBP1 mediates RIPK3-dependent cell death and MAVS-dependent pathogenic type I IFN response [50]. Further, Caspase-8-dependent apoptosis also contributes to the disease under ADAR1 deficiency, which is induced by the constitutive combination of ZBP1 and RIPK1 [51]. Caspase-8 also inhibits the ZBP1-mediated nuclear factor-kappaB (NF-κB) inflammatory pathway. Further investigations suggested that endogenous Alu dsRNA may be the ligand recognized by ZBP1 in the case of ADAR1 deficiency [52]. Nonetheless, a related study also identified and confirmed a small molecule, CBL0137, which promoted the utilization of Z-DNA conformation by the genome sequence [51]. Therefore, CBL0137 generates a large amount of endogenous Z-DNA and induces ZBP1-dependent cell death in tumor stromal fibroblasts during ADAR1 inhibition.

Both ADAR1 and ZBP1 are induced by IFN, but ADAR1-P150, one of its subtypes, can inhibit the function of ZBP1. ADAR1-P150 attenuates the synthesis of endogenous Z-RNA in the nucleus and inhibits the recognition of Z-NA of ZBP1 by combining with it in the cytoplasm. A small molecule drug CBL0137 promotes the synthesis of endogenous Z-DNA in the nucleus and plays an important role in inducing the ZBP1-mediated signal pathway. When ADAR1 is defective, ZBP1 mainly causes two forms of cell death: necroptosis, and apoptosis, which depend on the recognition of the Zα2 domain. Necroptosis is mainly caused by the ZBP1-mediated activation of the RIPK3-MLKL signal axis, while apoptosis is caused by the constitutive combination of ZBP1 and RIPK1 to induce the activation of Caspase-8. Caspase-8 can also inhibit the effects of ZBP1 and RIPK3 to inhibit necroptosis. In addition, ZBP1 also promotes type I IFN responses by inducing the mitochondrial antiviral-signaling (MAVS) pathway.

## 3. ZBP1 Mediates Necroptosis

In previous studies, necrosis was considered to be a passive and unregulated form of cell death [3,53,54]. However, in recent years, a special form of programmed cell death, namely necroptosis, has been reported [55,56,57,58]. It is characterized by necrotic death and is also regulated by related molecules, including RIPK1/3 [59,60,61,62]. This kind of programmed cell death can be induced by multiple factors, including TNF, IFN, LPS, dsRNA, DNA damage, and endoplasmic reticulum stress [63,64].

Necroptosis is caused by a combination of different ligands with TNF family death domain receptors, pattern recognition receptors, and virus sensors via an independent and unified downstream pathway [65,66,67]. TNF-induced necroptosis is the most studied and classic pathway, which is mediated by RIPK1, RIPK3, and MLKL [68,69,70]. TNF binds to the corresponding receptor (TNFR1), and its death domain TRADD binds and activates RIPK1. In the absence of Caspase-8, FADD is further recruited to form a complex, which acts on RIPK3 to activate phosphorylation and oligomerization [71,72,73,74]. Finally, the necrosome composed of RIPK3 activates the MLKL protein. MLKL is activated by phosphorylation at different sites in different species [75,76,77]. Human MLKL is phosphorylated at Thr357, Ser358, Ser345, and Ser347, whereas mouse MLKL is phosphorylated at Thr349 and Ser352 [78]. As an executor, MLKL changes its conformation after activation via RIPK3 phosphorylation. It releases four helical bundle domains, followed by translocation from the cytoplasmic matrix to the cell membrane, leading to structural disintegration of the plasma membrane [64,79,80]. The leaked cellular components may bind to the original and surrounding cells as ligands to further induce necroptosis.

ZBP1 is the master regulator of one of the induction pathways of necroptosis, which is mainly caused by virus infection [81]. It is associated with the induction and execution of necroptosis. The biggest difference between this pathway and the classical pathway lies in the role played by RIPK1, which often exists as a negative regulator of necroptosis in the ZBP1-mediated pathway [21,27,82]. RIPK3 and MLKL mediate the signal transduction at the final stages of necroptosis by integrating different signals to determine the extent of necrosis.

### 3.1. ZBP1 Interacts with Key Molecules in Necroptosis Signal Transduction via RHIM Domain

The signal transduction of necroptosis involves four proteins carrying RHIM domains, namely ZBP1, RIPK1, RIPK3, and TRIF [83]. The role of TIR domain-containing adaptor inducing interferon-β (TRIF) is similar to that of ZBP1 in necroptosis. As an adapter of Toll-like receiver 3/4 (TLR3/4), it interacts with RIPK3 to mediate necroptosis [84]. ZBP1 is associated with another initiating pathway, which induces necroptosis by combining the RHIM domain with RIPK3. RIPK1 also regulates this process via the RHIM domain.

#### 3.1.1. ZBP1 Combines with RIPK3 during the Formation of Necrosome

The necrosome was first proposed in the typical necroptosis pathway induced by TNF [85]. It is a cytoplasmic amyloid complex, mainly composed of activated RIPK1, RIPK3, and MLKL, which trigger necroptosis [86]. The core function of the necrosome is to promote the recruitment and phosphorylation of RIPK3 and MLKL. In the TNF pathway, RIPK1 promotes the autophosphorylation and activation of RIPK3. While in ZBP1-mediated necroptosis, ZBP1 induces the autophosphorylation of RIPK3 (Figure 3). The interaction between ZBP1 and RIPK3 is also sufficient to generate another type of necrosome and activate MLKL. RIPK1 plays the opposite role in this pathway and inhibits necroptosis. During mouse development, the deletion of RIPK1 induces ZBP1-mediated necroptosis and apoptosis, resulting in perinatal death [27,82,87]. The loss of keratinocyte RIPK1 triggers skin inflammation and necroptosis [21]. RIPK1 has no kinase activity without induction of TNF and other factors. However, it can bind to RIPK3 via the RHIM domain, and it cannot promote RIPK3 phosphorylation. In this case, other proteins that activate necroptosis, such as TRIF and ZBP1, cannot bind to RIPK3, suggesting that RIPK1 inhibits ZBP1-mediated necroptosis.

When viruses or endogenous Z-NA are accumulated, ZBP1 plays a critical role in the induction of necroptosis. After its Zα2 domain sensed Z-NA, ZBP1 can phosphorylate and activate RIPK3 by directly binding to it, which depends on their RHIM domains. The activated RIPK3 spontaneously oligomerizes to form necrosomes and induces activation and oligomerization of MLKL to carry out necroptosis. Therefore, the function depending on this domain is inhibited by other proteins with the RHIM domain, including RIPK1 and M45 protein in MCMV. In addition, LPS produced in other pathogenic infections can also recognize TLR4 receptors on the cell membrane to induce the activation of RIPK3 to form necrosomes, and the connection between this receptor and RIPK3 is also realized by the protein with RHIM domain, TRIF. Another classic pathway of necroptosis is mediated by TNF, which can recognize more abundant pathogenic signals. Excessive TNF binds to TNFR, which can combine with FADD and RIPK1 to form a complex which activates RIPK3 to promote the formation of necrosomes.

#### 3.1.2. Combination of ZBP1 and RIPK1

Both RIPK1 and ZBP1 contain RHIM domains, suggesting direct interaction [88]. ZBP1, as an RHIM protein, not only participates in necroptosis but also regulates apoptosis using Caspase-8 as the main executor by controlling the formation of a complex called TRIFosome [42]. TRIFosome is composed of ZBP1, RIPK1, FADD, and Caspase-8. In the case of LPS induction, TLR4 recruits RIPK1 via TRIF bound to ZBP1, resulting in the assembly of TRIFosome, followed by the activation of Caspase-8, resulting in apoptosis [34]. In addition, the formation of this complex is also crucial to inflammasome activation. In another study, the interaction between ZBP1 and RIPK1 also activated the NF-κB pathway [26], which led to the activation of type I IFN and other cytokines.

### 3.2. ZBP1 Mediates the Activation of NLRP3 Inflammasome to Induce PANoptosis

NLRP3 is a typical inflammasome sensor, which initiates the assembly of the inflammasome in the innate immune system [53,89]. NLRP3 inflammasome mediates the activation of Caspase-1 to induce pyroptosis during IAV infection, which is a typical inflammatory cell death [81]. NLRP3 recruits and activates Caspase-1 via apoptosis-associated speck-like protein containing a CARD (ASC) carrying the Caspase recruitment domain. The activated Caspase-1 enables the pro-inflammatory factor pro–IL-1 by cleaving β pro-IL-18, resulting in the activation of pro-IL-18. The N-terminal of the pole-forming protein gasdermin D (GSDMD) is released, which simultaneously induces a pro-inflammatory reaction and pyroptosis [90]. The upstream of NLRP3 is regulated by the RIPK1-RIPK3-Caspase-8 axis mediated by ZBP1 [91].

However, the role of NLRP3 inflammasome mediated by ZBP1—extends to PANoptosis, which was first proposed in 2019 to describe cell death and inflammation caused by IAV infection (Figure 4) [92]. PANoptosis represents a combination of pyroptosis, apoptosis, and necroptosis, which are common regulatory mechanisms with mutual crosstalk [93,94]. The three play a redundant role in the initiation and amplification of inflammation [95]. When one of the pathways is blocked, an antiviral inflammatory reaction can still occur. ZBP participates in PANoptosis by triggering the assembly of a signal conduction complex. ZBP1-NLRP3 inflammasome may be assembled with ZBP1-RIPK3-FADD-cassase-8 complex to form large multi-protein complexes constituting a PANoptosome [96]. This complex is involved in NLRP3 inflammasome-dependent pyroptosis, Caspase-8-mediated apoptosis, and MLKL-driven necroptosis [97]. This concept has been extended from IAV to HSV1, coronavirus [98], Fungi [41], *Yersinia* [99], tumor [100], and nerve injury [101] and is under continuous development.

PANoptosis represents a combination of pyroptosis, apoptosis, and necroptosis, which is mediated by ZBP1 following IAV infection and other viral infections and inflammation. After sensing a large amount of IAV Z-RNA, ZBP1 can combine with RIPK1, RIPK3, Caspase-8, FADD, NLRP3, ASC, and Caspase-1 to form a giant complex called the PANoptosome. Among these molecules, RIPK1, RIPK3, FADD and caspase-8 are related to apoptosis. The activation of the molecules ultimately induces activation of caspase-8, which acts on the executor Caspase-3/6/7 and leads to apoptosis. While RIPK3 is mainly related to necroptosis, RIPK1 and FADD are also considered to play a positive role in the occurrence of necroptosis. The activation of RIPK3 directly activates and oligomerizes MLKL, the executor of necroptosis, to form an ion channel that destroys the plasma membrane. NLRP3, ASC, and Caspase-1 are key molecules in the occurrence of pyroptosis. They can form NLRP3 inflammasomes to promote the generation of final executors of pyroptosis. NLRP3 is responsible for sensing the corresponding stimulus. ASC has a PYD domain and a CEAD domain that can be recruited by NLRP3 and then recruit Caspase-1. Caspase-1 cleaves and activates the final executor, GSDMD. Pyroptosis is mainly caused by the N-terminal domain of GSDMD, which can transfer to the cell membrane and promote pore formation, leading to the release of pro-inflammatory cytokines IL-1 β and IL-18.

## 4. The Role of ZBP1 in Human Diseases

In human diseases caused by viruses, such as influenza and smallpox, ZBP1-mediated signal pathways control the programmed death of infected cells and related inflammatory responses [37,102]. In addition, ZBP1 also plays an important role in regulating necroptosis in other human diseases, such as cancer and systemic inflammatory disease (Table 1).

### 4.1. ZBP1 as a Sensor of IAV-Induced Necroptosis

IAV is an antisense RNA virus belonging to the family *Orthomyxoviridae*, which causes lung damage and related diseases in infected mammals [108]. In recent years, studies involving human diseases by ZBP1 focused on lung loss caused by IAV infection. Meanwhile, studies involving mouse cells infected by IAV also revealed various upstream and downstream mechanisms of ZBP1-mediated necroptosis [33]. The possibility of ZBP1 as a cytoplasmic DNA sensor has been proposed for a long time. By 2016, relevant studies established that ZBP1 was a congenital sensor of IAV, and ZBP1 sensed IAV genomic RNA to activate RIPK3 [26]. During IAV infection, the role of ZBP1 sensing was mediated by the combination of polymerase subunit PB1 and nucleoprotein NP. In a related study in 2017 [35], it was suggested that ZBP1 recognized a vRNP complex, which is composed of an IAV RNA genome, multiple NPs, and PBs. The ZBP1 activation may also require RIG-I signal transduction and ubiquitination. Nonetheless, the Zα2 domain of ZBP1 plays a key role in signal transduction by directly binding with Z-NA. In the study of IAV, several molecules regulate ZBP1-induced cell death in different ways, including IRF1 [109], Caspase-6 [110], and TRIM34 [111]. IFN regulatory factor (IRF) 1 is a molecule that upregulates ZBP1 transcription. However, in mouse cells infected with IAV, IRF1 alone cannot alter the cell death and inflammatory response caused by ZBP1, perhaps because it is only one of the factors affecting ZBP1 expression, and its role can be replaced by other similar molecules. 

Caspase-6 was considered an executor caspase, which plays a role in the execution of apoptosis [112]. However, the study found that Caspase-6 can promote three major types of programmed cell death in IAV infection by binding to RIPK3 and strengthening the formation of the PANoptosis complex. TRIM34 is a member of the tripartite motif (TRIM) family [113]. Many TRIM family members exhibit E3 ubiquitin ligase activity [114]. It is related to the polyubiquitination of K63 at the K17 residue of ZBP1, which promotes the combination of ZBP1 and RIPK3. 

From another perspective, the study of ZBP1 in IAV infection revealed the kind of RNA recognized by ZBP1. The above studies also indicated that short IAV gene fragments might be used as ligands for ZBP1 recognition. Accordingly, a study in 2020 reported that IAV generated Z-RNA via its DVG for ZBP1 [37]. After IAV infection, the genomic RNA entered the nucleus of the host to promote self-replication, in addition to the activation of necroptosis in the nucleus, which is different from the classic TNF-α activated pathway occurring in the cytoplasm. The “defective interference” (DI) particles formed by DVG packaging carrying higher concentrations of DVG RNA can trigger rapid phosphorylation of MLKL. The use of anti-Z-NA serum can obviously stain the nucleus during the previous infection. In this process, ZBP1 is recruited from the cytoplasm to the nucleus. MLKL, the executor of necroptosis, is also located in the nucleus and mediates the rupture of the nuclear membrane independent of apoptosis. The subsequent release of nuclear DAMPs promotes neutrophil recruitment and activation, which aggravates IAV infection symptoms. The specific mechanism of IAV-inducing necroptosis was also verified in other *Orthomyxoviridae* families, demonstrating the core function of ZBP1 in sensing Z-NA-mediated necroptosis [38].

### 4.2. ZBP1-Dependent Inflammatory Cell Death in Coronavirus Infection

Coronavirus has received wide attention following the outbreak in 2019 [115,116]. Coronavirus is a single-stranded positive RNA virus, which can be classified into seven types: 2019 nCoV, HCoV-229E, HCoV-OC43, HCoV-NL63, HCoV HKU1, SARS CoV, and MERS CoV [117]. Among them, SARS CoV-2 infection causes respiratory inflammation in the host but also nerve damage, resulting in a variety of nervous system complications [118,119]. However, as early as 2017, it was found that human coronavirus induces necroptosis of human nerve cells [120]. HCoV-OC43 strain infects mice and induces nerve cell death in large numbers depending on RIPK1/3 and MLKL via necroptosis. The induction of cell death, also found in the mouse hepatitis virus (MHV), which is homologous to coronavirus in mice, even drives the inflammatory reaction and cell death with PANoptosis as the core [98]. 

It also demonstrates that the concept of PANoptosis is widely applicable to the study of virus infection. Transgenic mice (K18-hACE2) expressing human angiotensin-converting enzyme 2 under the cytokeratin 18 promoter are widely used to study the pathogenesis of SARS CoV-2 infection [121]. The neural cell culture line of K18-hACE2 and the brain after SARS CoV-2 infection showed upregulation of inflammation-related genes. In addition, the protein and mRNA levels of ZBP1 and pMLKL also increased 1 to 3 days after infection, which directly demonstrated that ZBP1 induced by SARS CoV-2 mediates the occurrence of necroptosis [122]. IFN therapy for SARS CoV-2 has limited value and even negative effects [19]. The main reason is that the treatment with exogenous IFN enhanced the ZBP1-mediated PANoptosis and cytokine storm during coronavirus infection, leading to lung injury and even individual death. This study also found that the high expression of ZBP1 and IFN often occurred in critically ill patients with COVID-19, suggesting that these molecules play a negative role in disease treatment. This also provides a strategy for combination therapy by blocking ZBP1 during IFN therapy. 

Specific molecules regulate ZBP1 sensing of Z-NA-mediated necroptosis in coronavirus, which may be attributed to the co-evolution of the virus and host immune defense system. Non-structural protein 13 (Nsp13) existing in SARS CoV exhibits this function. Nsp13 is a helicase and carries a potential RHIM domain [123]. It may inhibit ZBP1-mediated cell death by preventing the formation of Z-RNA and inhibiting the interaction between ZBP1 and RIPK3. Altogether, ZBP1-dependent cell death and inflammatory response are of positive or negative significance in diseases caused by coronavirus infection. The study of ZBP1-mediated PANoptosis may provide important theoretical support for SARS remission and treatment.

### 4.3. Vaccinia Virus Inhibits ZBP1-Mediated Necroptosis

VACV is a poxvirus, which is a double-stranded DNA virus [124]. It is closely related to smallpox and cowpox viruses in immunity and can be used as a vaccine against smallpox. VACV exhibits immune escape, mediated via nearly a third of its genes. One of the main escape genes, E3L, encodes the E3 protein. E3 has a double-stranded RNA (dsRNA)-binding domain at the C-terminal and a Z nucleic acid-binding domain at the N-terminal [125]. The C-terminal domain has been shown to inhibit IFN-induced activation of dsRNA-dependent antiviral enzymes. The N-terminal Zα domain is related to ZBP1-mediated necroptosis [24]. In this study, WT type VACV and VACV-E3L Δ 83N with deleted Zα domain of E3 was used to infect IFN-treated mouse L929 cells to explore the role and mechanism of the E3 N-terminal, which demonstrated its role in the inhibition of IFN activity. 

Cells infected with the E3-deficient virus showed RIPK3-dependent necroptosis, while the E3 N-terminal Zα domain competed with ZBP1 to prevent ZBP1-dependent activation of RIPK3 in VACV-infected cells. Further, VACV only inhibited ZBP1-mediated necroptosis but not RIPK1-mediated necroptosis in the TNF-induced pathway. Regarding the inhibition of necroptosis, other strategies have also been discovered in poxvirus [126]. The virus MLKL protein derived from BeAn 58,058 and *Cotia* poxvirus blocked the activation of MLKL and necroptosis in cells by isolating RIPK3. The study of VACV, or the whole poxvirus, is of great significance in screening for inhibitors of necroptosis.

### 4.4. Heat Stress Activates ZBP1 via Z-NA-Independent Mechanism in Heat Stroke

Heat stroke is a disease associated with high body temperature and metabolic disorders mainly caused by heat stress [127]. In severe cases, systemic inflammatory reactions and multiple organ failure may occur, resulting in death. Here, we specifically discuss the role of ZBP1 in this disease because the latest related study in 2022 [104] reported a unique mechanism of necroptosis. The study first demonstrated that heat stress induces cell death as well as other inflammatory reactions via RIPK3-dependent activation of MLKL and Caspase-8 in mice and L929 cells, resulting in pathological manifestations of heat stroke. In ZBP1-defective cells but not deficient in other RIPK3 interacting proteins, signs associated with all kinds of cell death induced by heat stress disappeared, such as the phosphorylation of RIPK3 and MLKL, which was different from heat stress in normal cells. Thus, ZBP1 is a key molecule associated with RIPK3-mediated cell death in heat stress. In human HT-29 cell lines expressing RIPK3 and RIPK1 but not ZBP1, heat stress did not induce cell death. 

However, the application of exogenous human ZBP1 increased its sensitivity to heat stress-induced cell death, which further demonstrates that ZBP1 plays a key role in heat stress-induced cell death. Heat shock transcription factor 1 (HSF1), as a regulatory molecule in heat stress, has been found to be a key factor in promoting the expression of ZBP1 in heat stress [128]. Notably, the increase in ZBP1 expression alone is not enough for cell death. In heat stress, ZBP1 activation occurred via a Z-NA-independent mechanism, which may be related to its dependence on the RHIM domain for aggregation. This study undoubtedly provides insight into the role of ZBP1. The activation of ZBP1 and induction of cell death do not necessarily require the detection of pathogens or endogenous Z-NA. Certainly, this unique mechanism requires further study. Among various pathogenic infections, high fever is also a common symptom, in which heat stress may eliminate pathogens by activating ZBP1 to promote cell death. However, excessive heat stress also adversely affects the organism.

### 4.5. Other Diseases

ZBP1, as a key regulatory molecule of cell death and inflammation, plays a role in many human diseases in addition to the aforementioned ones. Human Cytomegalovirus (HCMV) infection causes visceral diseases. ZBP1-induced IRF3 transcription and IFN-β expression. The overexpression of ZBP1 inhibits HCMV replication [105]. In chronic airway inflammation caused by smoking, ZBP1 induces inflammation by binding to damaged mitochondrial DNA (mtDNA) released into the cytoplasm under oxidative stress [29].

Another important human disease related to ZBP1 is cancer. ZBP1 plays a key role in different stages of tumors and may be a therapeutic target [129]. During the development of solid tumors, necroptosis may occur in the core region, which is called tumor necroptosis, which may cause tumor metastasis [130]. Studies based on MVT-1 breast cancer models demonstrated that ZBP1, rather than RIPK1, mediates tumor necroptosis [20]. The strong expression of ZBP1 and RIPK3 in necroptosis was also verified in other types of solid tumors. In addition, tumor necroptosis is most likely caused by glucose deprivation (GD) and may be mediated via mtDNA, which is released by stress under the regulation of GD and recognized by the Zα domain of ZBP. The anti-tumor efficacy of radiotherapy may be related to the relationship between ZBP1-mediated necroptosis and the stimulator of the interferon genes (STING) pathway in tumors [106]. The inhibitory effects of radiotherapy on tumor growth in the MC38 mouse colon adenocarcinoma cell line and B16-SEY mouse melanoma cell line are directly related to the expression of MLKL in tumor cells, via ZBP1-mediated necroptosis signal transduction. Further, during radiotherapy, ZBP1-MLKL necroptosis promotes STING activation and type I IFN response in tumor cells accumulating cytoplasmic mtDNA. ZBP1-mediated necroptosis can be enhanced via Caspase-8 gene ablation in tumor cells to improve the effects of radiotherapy. Fisetin is a natural flavonoid routinely used to inhibit the development of cancer. It promoted the death of human ovarian cancer cell lines via ZBP1-mediated necroptosis and other mechanisms [107]. However, the mechanism of fisetin-induced cell death and its application require further investigation. 

ZBP1-mediated cell death and other intracellular signal pathways also occur in neurodegenerative diseases, a variety of inflammations, fungal, bacterial, and *T. gondii* infections, and other pathologies. All kinds of diseases are related to necroptosis, suggesting the need for identifying its mechanism in different pathologies.

## 5. ZBP1 Regulation and Prospects

During ZBP1 regulation, recent studies have identified several important molecules which could affect the function of ZBP1 in different aspects. At the transcription level, IRF1 and HSF1 regulate ZBP1 and thus promote the expression of ZBP1. TRF3-Thr-AGT decreases ZBP1. IRF1 is a member of the IRF family of transcription factors and was first identified as the transcription activator of the IFN and IFN-stimulated gene (ISG) [131]. In IRF1 deficient cells infected with IAV, the expression level of ZBP1 was downregulated, which was also confirmed in a variety of cells and under different stimulation conditions [109]. The regulatory effect of HSF1 on ZBP1 is the same as that described previously [104]. There was an HSF1 binding site in the promoter region of ZBP1, and the deletion of this site or HSF1 inhibited the increase in ZBP1 expression induced by heat stress. Endogenous transfer RNA (tRNA) is a kind of non-coding RNA, and its derived small RNA (tsRNA) is related to many diseases [132,133]. The TRF3-Thr-AGT screened from them had been proven to be closely related to acute pancreatitis (AP) development. Bioinformatics predicts that TRF3-Thr-AGT can bind to the 3′ untranslated regions (3′UTR) of ZBP1. The experiment also proved that the inhibition of TRF3-Thr-AGT overexpression on cell death in the AP model could be eliminated by upregulating ZBP1 [134]. It suggests that tRF3-Thr-AGT inhibits cell death and inflammation by inactivating the ZBP1/NLRP3 pathway.

Caspase-6, TRIM34, Pyrin, AIM2, and ABT-737 promote cell death via enhanced interaction between ZBP1 and RIPK3. By contrast, MCMV M45 [135], IE3 [136], VZV ORF20 [137], VACV E3 [24,103,138], herpes simplex virus type 1(HSV1) ICP6 [139,140], and RIPK1 [21,141,142] mostly carry RHIM domains, which combine with ZBP1 and RIPK3. Under IAV infection, Caspase-6 can combine with RIPK3 to strengthen the formation of PANoptosome, and both the large and small aggregates of Caspase-6 are critical for the binding of RIPK3 to ZBP1 [110]. The association between TRIM34 and ZBP1 promotes ZBP1 recruitment of RIPK3, and TRIM34 mediates K63-linked polyubiquitination of ZBP1 at residue K17 [111]. Absent in melanoma 2 (AIM2) is a member of the Pyrin and HIN domain protein family, which can recognize double-stranded DNA to form an inflammasome. In HSV1 and F. novicida infection, AIM2, Pyrin, and ZBP1 together with ASC, Caspase-1, Caspase-8, RIPK3, RIPK1, and FADD form a large multi-protein complex called AIM2 PANoptosome, which drives PANoptosis [96]. ABT-737 is a Bcl-2 homology 3-mimetic drug. In bladder cancer cells, ABT-737 induces cell necrosis when either ZBP1 or RIPK3 are knocked down, which is achieved by upregulating the interaction between ZBP1 and RIPK3 [143]. The molecules that can inhibit the interaction of ZBP1 and RIPK3 mostly exist in viruses and have RHIM domains, which may be the result of the co-evolution of the virus and host immune defense [144]. Additionally, RIPK1, as a molecule inducing necroptosis in most cases, can competitively combine with RIPK3 in development and endogenous Z-NA-mediated necroptosis to play an inhibitory role. 

Several molecules also indirectly regulate ZBP1. PUMA can be induced by necroptosis and activates ZBP1 sensation by promoting mtDNA release [145]. Nonylphenol (NP) reduces the degree of ZBP1 promoter methylation and promotes ZBP1 expression by inhibiting the binding of LncRNA PVT1, EZH2, DNMT1, and ZBP1 promoter region [146]. CBL0137 activates ZBP1-mediated necroptosis by promoting Z-DNA synthesis. The discovery of additional regulatory molecules in different diseases related to ZBP1 and identified pathogens is also a core research strategy [47]. However, chemical inhibitors that directly affect ZBP1 are currently unavailable.

## 6. Conclusions

Studies investigating ZBP1 originated in its Zα and RHIM domains, which interact with other molecules upstream or downstream during signal transmission. Currently, studies suggest that ZBP1 recognizes Z-NA, mediated directly by its second Zα domain at the N-terminal. Further, necroptosis is the most studied ZBP1-mediated pathway. Although ZBP1-mediated necroptosis is not the most classical pathway, ZBP1-induced necroptosis via the RIPK3-MLKL axis has been established in a variety of human diseases, indicating that ZBP1 may be a potential therapeutic target.

Analysis of the classical role of ZBP1 in viral infection is also related to its original role as a viral sensor. In IAV studies, the vRNA-mediated by DVG generated RNP and was identified by ZBP1. In addition, endogenous nucleic acids were recognized by ZBP1. MtDNA [29] and dsRNA [38] from ERE may cause a variety of chronic inflammations via ZBP1-mediated immune defense mechanisms. In the future, the role of ZBP1 in different virus infections needs to be explored to determine the genome sequence that produces Z-NA.

ZBP1 mediates other cell death pathways, such as apoptosis, pyroptosis, and PANoptosis, which integrates the former two and necroptosis. It is also a focus of current and future research, including SARS-CoV-2 infection and the control of tumors. It is worthwhile to explore the mechanism of ZBP1 in different diseases. 

In terms of the regulation of ZBP1, existing studies have found that many molecules can affect the function of ZBP1 at the transcriptional level, its interaction with its protein, and indirectly. It is of great significance to continue to search for more related molecules in these areas and explore molecules that can affect the action of ZBP1 in other ways. In addition, there is a lack of small molecular substances that can be synthesized in vitro and directly affect ZBP1-mediated cell death function in relevant fields at present, which is what we will actively look for in the future.

## Figures and Tables

**Figure 1 molecules-28-00052-f001:**
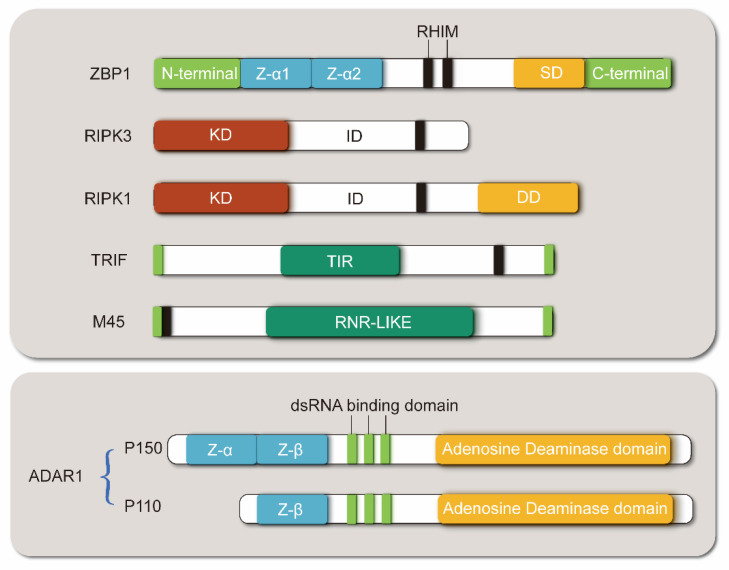
Structural diagram of ZBP1 and its interacting proteins.

**Figure 2 molecules-28-00052-f002:**
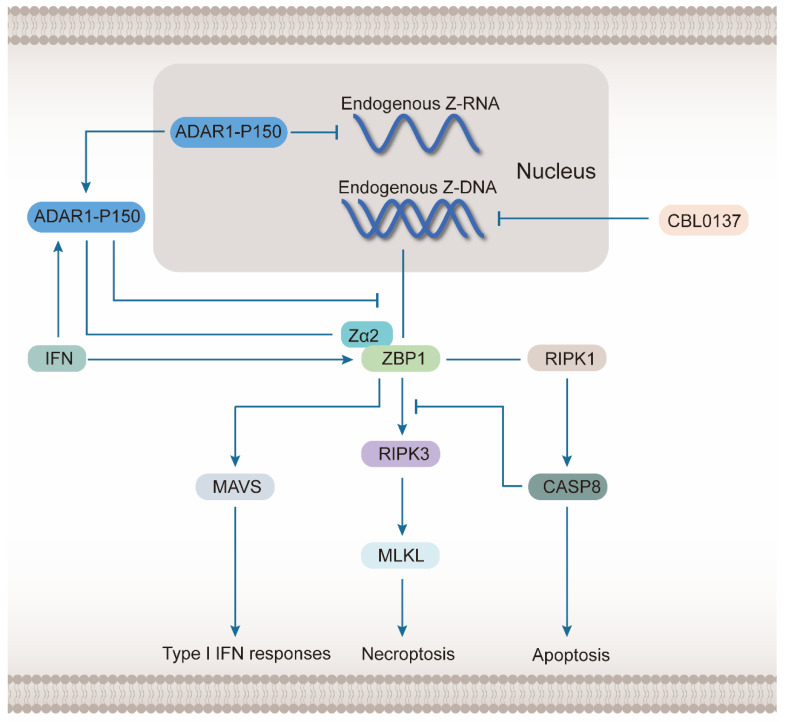
ADAR1-P150 inhibits ZBP1-mediated programmed cell death and inflammation.

**Figure 3 molecules-28-00052-f003:**
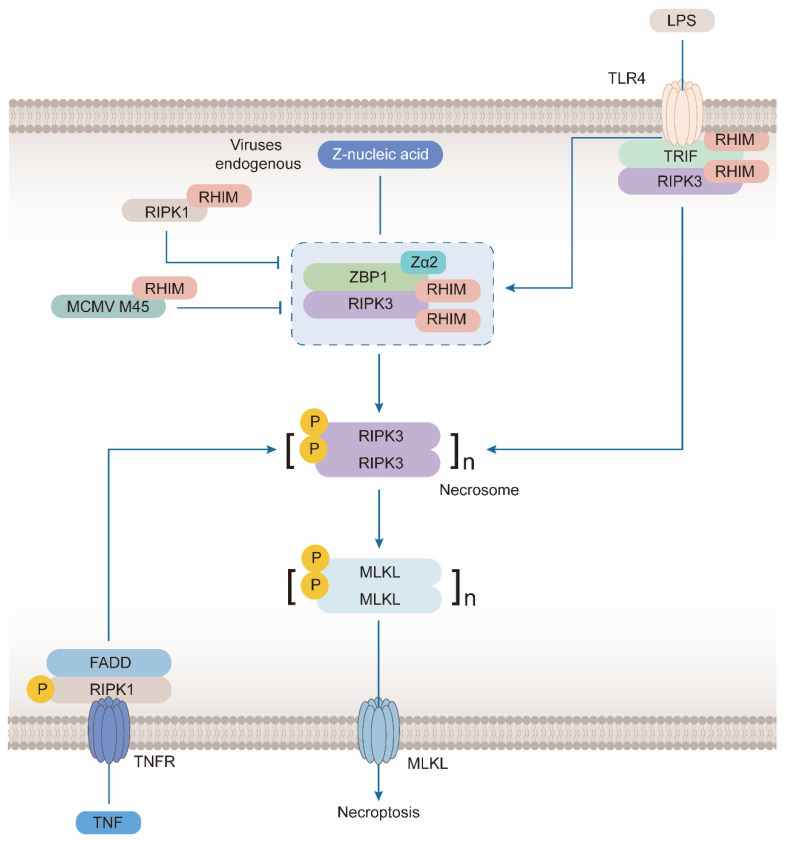
ZBP1 induces formation of necrosomes in Necroptosis.

**Figure 4 molecules-28-00052-f004:**
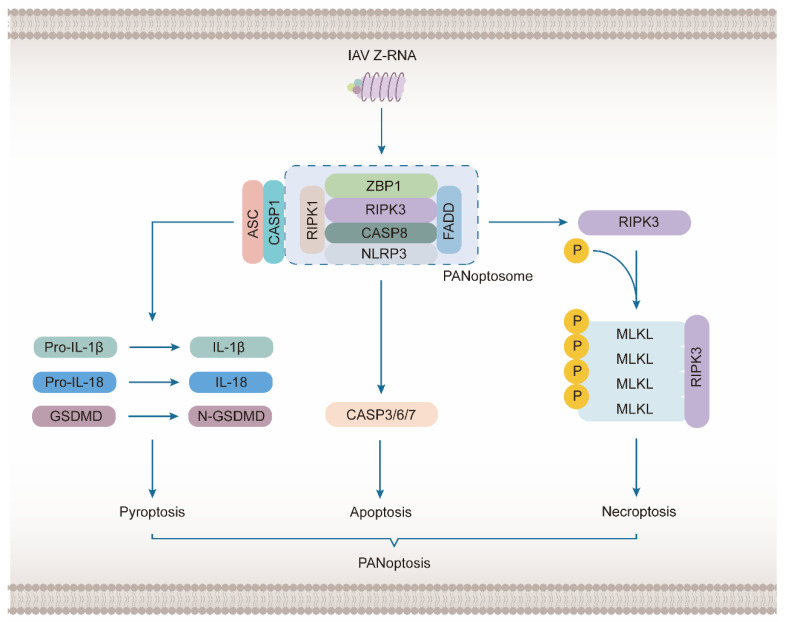
ZBP1 Induces PANoptosis following IAV Infection.

**Table 1 molecules-28-00052-t001:** ZBP1 mediates cell death and inflammation in different diseases.

Diseases	Factors	Major Finding	References
Injury caused by IAV infection	IAV	Replicating IAV generates Z-RNAs, which activate ZBP1 in the nucleus of infected cells	[37]
SARS	SARS-CoV-2	ZBP1 induced during coronavirus infection limits the efficacy of IFN therapy by driving inflammatory cell death and lethality	[19]
Smallpox	VACV	Zα-deficient E3 dsRBD promotes formation of Z-RNA and recruitment of ZBP1	[103]
Heatstroke	Heat stress	ZBP1 plays a mediating role in heat stress-induced cell death in the form of aggregation independent of its Zα Domain	[104]
Splanchnic disease	HCMV	ZBP1 was found to be able to induce IRF3 transcription and IFN-β	[105]
Chronic airway inflammation	mtDNA	ZBP1 combines with damaged mtDNA released into the cytoplasm due to oxidative stress to induce inflammation	[29]
Cancer	Radiation	A previously unrecognized crosstalk between ZBP1-MLKL necroptotic cascade and STING-mediated cytosolic DNA sensing	[106]
Ovarian cancer	Fisetin	Fisetin-induced OC cell death involves apoptosis and necroptosis, while ZBP1 regulates necroptosis through RIP3/MLKL pathway	[107]
Systemic inflammatory disease	IFN	ZBP1 induced by IFN-γ via the JAK1/STAT1 signaling pathway, is necessary for IFN-γ-induced necroptosis	[52]

## Data Availability

Not applicable.

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
