# Peer review of "ZBP1-Mediated Necroptosis: Mechanisms and Therapeutic Implications"

_molecules, 2022, doi:10.3390/molecules28010052_

Round 1

Reviewer 1 Report

The manuscript “ZBP1-mediated Necroptosis: Mechanisms and Therapeutic Implications” reviews the recent developments related to the function of ZBP1 protein and its role in necroptosis.

The study of necroptosis is an area of active development with a potential for development of therapeutic products. In the wake of discovery of the role of ZBP1 in viral infections including SARS-CoV-2 the review is well merited.

Manuscript is well written and except few minor notes is sufficient for publishing. Few notes below:

-          As common in the field, the review is full of abbreviations. Considering that Molecules is a journal aimed at a wider audience of chemists, I strongly advice adding a list of abbreviations for the reader comfort.

-          The section in conclusions on lines 436 to 450 describes the effects of different molecules on the action of ZBP1. In my opinion this information is important especially for chemists involved in the drug design and development, and must be expanded as a separate chapter in the discussion section.

-          Most references are in bold script, but some are not, e.g., on lines 436 to 446.

Author Response

Comments and Suggestions for Authors

The manuscript “ZBP1-mediated Necroptosis: Mechanisms and Therapeutic Implications” reviews the recent developments related to the function of ZBP1 protein and its role in necroptosis. The study of necroptosis is an area of active development with a potential for development of therapeutic products. In the wake of discovery of the role of ZBP1 in viral infections including SARS-CoV-2 the review is well merited. Manuscript is well written and except few minor notes is sufficient for publishing. Few notes below.

Response: We are grateful to receive your comments. Thank you for your high evaluation of our article.

As common in the field, the review is full of abbreviations. Considering that Molecules is a journal aimed at a wider audience of chemists, I strongly advice adding a list of abbreviations for the reader comfort.

Response: Thank for your valuable advice, and we have added a list of abbreviations (please see lines 542-564).

The section in conclusions on lines 436 to 450 describes the effects of different molecules on the action of ZBP1. In my opinion this information is important especially for chemists involved in the drug design and development, and must be expanded as a separate chapter in the discussion section.

Response: According your advice, we have added a separate section in the revised manuscript, which describes the effects of different molecules on the action of ZBP1, including the expression level of ZBP1, the direct combination of ZBP1, and indirect influence (please see section “5. ZBP1 REGULATION AND PROSPECTS”, lines 469-514).

Most references are in bold script, but some are not, e.g., on lines 436 to 446.

Response: Thank for your reminder. We have unified the citation format in the revised manuscript. Thanks again for your kind help.

Reviewer 2 Report

The manuscript compiles a number of studies on ZBP1 and its possible biological functions. While it may be good information for specialists in the field, it is hard work for any non-specialist researcher to follow. I also miss, given the audience of the Journal, a greater detail in the molecular aspects of the described interactions. I encourage the authors to improve their writing to reach a wider audience.

1. What is the main question addressed by the research?
IT IS A REVIEW ON THE ROLE OF ZBP1 PROTEIN ON NECROPTOSIS
2. Do you consider the topic original or relevant in the field?
YES
Does it address a specific gap in the field?
YES, IT DOES AS A REVIEW
3. What does it add to the subject area compared with other published
material?
IT REVIEWS THE PARTICULAR AND RELEVANT ROLE OF ZBP1 ON NECROSOME FORMATION
4. What specific improvements should the authors consider regarding the
methodology? What further controls should be considered?
THIS QUESTION DOES NOT APPLY, IT IS A REVIEW ARTICLE
5. Are the conclusions consistent with the evidence and arguments presented
and do they address the main question posed?
YES, BUT THEY CAN BE IMPROVED
6. Are the references appropriate?
YES
7. Please include any additional comments on the tables and figures.
FIGURES CAN BE ALSO IMPROVED TO PROVIDE MORE DETAILS 

Author Response

Comments and Suggestions for Authors

The manuscript compiles a number of studies on ZBP1 and its possible biological functions. While it may be good information for specialists in the field, it is hard work for any non-specialist researcher to follow. I also miss, given the audience of the Journal, a greater detail in the molecular aspects of the described interactions. I encourage the authors to improve their writing to reach a wider audience.

Response: We are grateful to receive your comments. We have introduced ZBP1 in more detail way, and briefly described different types of programmed cell death, such as the apoptosis, necroptosis and pyroptosis, involved with ZBP1 in the introduction (please see lines 30-55).

Programmed cell death plays an important role in life activities. Apoptosis is the first programmed cell death pathway discovered and studied. When apoptosis occurs, it shows cell volume shrinkage, chromatin agglutination, formation of apoptosome, and phagocytosis. The execution of this pathway is considered to be related to the Bcl-2 protein family and the Caspase family. Necroptosis is the path of passive death when cells are injured but procedural regulatory. This process depends on the phosphorylation of RIPK1, RIPK3 and MLKL. Pyroptosis is a new type of programmed death found in recent years, which is a typical inflammatory cell death. Inflammasomes play a major role in the pathogenesis and regulation, which activate Caspase family members to promote the activation of proinflammatory cytokines IL and gasfermin protein

Hope these explanations will facilitate readers' understanding.

  1. What is the main question addressed by the research?
    IT IS A REVIEW ON THE ROLE OF ZBP1 PROTEIN ON NECROPTOSIS
    2. Do you consider the topic original or relevant in the field?
    YES
    Does it address a specific gap in the field?
    YES, IT DOES AS A REVIEW
    3. What does it add to the subject area compared with other published
    material?
    IT REVIEWS THE PARTICULAR AND RELEVANT ROLE OF ZBP1 ON NECROSOME FORMATION
    4. What specific improvements should the authors consider regarding the
    methodology? What further controls should be considered?
    THIS QUESTION DOES NOT APPLY, IT IS A REVIEW ARTICLE
    5. Are the conclusions consistent with the evidence and arguments presented and do they address the main question posed?
    YES, BUT THEY CAN BE IMPROVED
    6. Are the references appropriate?
    YES
    7. Please include any additional comments on the tables and figures.
    FIGURES CAN BE ALSO IMPROVED TO PROVIDE MORE DETAILS 

Response: Your suggestions are very valuable. We have enhanced the figure legends by explain the domain information in each molecular in Figure 1 (please see lines 96-105), and supplement the details of molecular interactions in their respective pathways in Figures 2 (please see lines 169-180) and Figures 3 (please see lines 236-248), and provide a detailed description on the specific mechanisms of three kinds of programmed cell deaths included in PANoptosis in Figure 4 (please see lines 285-301). Thanks again for your kind help.

Reviewer 3 Report

 1. The introduction is quite inadequate since it does not provide the reader with a thorough knowledge of what ZBP1 is. It is worth noting that ZBP1 was the first cytosolic DNA sensor shown to work independently of TLRs. Furthermore, you should introduce the three types of cell death: necroptosis, apoptosis, and pyroptosis.

2. 3.1.1. ZBP1 combines with RIPK1 and RIPK3 during the formation of necrosome (lines 184-197), kindly consider revising this section because it is open to misinterpretation and is difficult to comprehend.

3. Please enhance the legends for figures 2, 3, and 4, and incorporate all information shown by the figure in the legend.

4. LPS should be stated as lipopolysaccharides prior to getting abbreviated, and mention the full name of DLM-1 and ADAR1 where they were first mentioned.

5. Please revise the entire manuscript and give adequate references that were missing (lines 136- 145, 169-174, 288-300, 330-338.

6. In line 256, “Besides, ZBP1 also plays an important role in regulating necroptosis in other human diseases, such as cancer and neurodegenerative diseases”, however, table 1 didn’t include any neurodegenerative disease.

Author Response

Comments and Suggestions for Authors

1.The introduction is quite inadequate since it does not provide the reader with a thorough knowledge of what ZBP1 is. It is worth noting that ZBP1 was the first cytosolic DNA sensor shown to work independently of TLRs. Furthermore, you should introduce the three types of cell death: necroptosis, apoptosis, and pyroptosis.

Response: We are grateful to receive your comments. According to your advice, we have introduced ZBP1 in more detail way, and briefly described different types of programmed cell death, such as the apoptosis, necroptosis and pyroptosis, involved with ZBP1 in the introduction (please see lines 30-55).

Programmed cell death plays an important role in life activities. Apoptosis is the first programmed cell death pathway discovered and studied. When apoptosis occurs, it shows cell volume shrinkage, chromatin agglutination, formation of apoptosome, and phagocytosis. The execution of this pathway is considered to be related to the Bcl-2 protein family and the Caspase family. Necroptosis is the path of passive death when cells are injured but procedural regulatory. This process depends on the phosphorylation of RIPK1, RIPK3 and MLKL. Pyroptosis is a new type of programmed death found in recent years, which is a typical inflammatory cell death. Inflammasomes play a major role in the pathogenesis and regulation, which activate Caspase family members to promote the activation of proinflammatory cytokines IL and gasfermin protein

Hope these explanations will facilitate readers' understanding.

  1. 3.1.1. ZBP1 combines with RIPK1 and RIPK3 during the formation of necrosome (lines 184-197), kindly consider revising this section because it is open to misinterpretation and is difficult to comprehend.

Response: Thank you for this suggestion. RIPK3 decisively forms necrosomes, which can be affected by many molecules. In ZBP1-related necroptosis, ZBP1 directly combines with RIPK3, which determines the occurrence of this process. We have revised the section by a more accurate description (please see section 3.1.1, lines 218).

  1. Please enhance the legends for figures 2, 3, and 4, and incorporate all information shown by the figure in the legend.

Response: Thank you for your kind advice. We have enhanced the figure legends by explain the domain information in each molecular in Figure 1 (please see lines 96-105), and supplement the details of molecular interactions in their respective pathways in Figures 2 (please see lines 169-180) and Figures 3 (please see lines 236-248), and provide a detailed description on the specific mechanisms of three kinds of programmed cell deaths included in PANoptosis in Figure 4 (please see lines 285-301).

  1. LPS should be stated as lipopolysaccharides prior to getting abbreviated, and mention the full name of DLM-1 and ADAR1 where they were first mentioned.

Response: Thank you for pointing this out. We have checked and corrected all abbreviations, and added a list of abbreviations blow the section of conclusion in the revised manuscript (please see lines 540-562).

  1. Please revise the entire manuscript and give adequate references that were missing (lines 136- 145, 169-174, 288-300, 330-338).

Response: Thank you for your suggestion. We have checked and added adequate references in the revised manuscript.

  1. In line 256, “Besides, ZBP1 also plays an important role in regulating necroptosis in other human diseases, such as cancer and neurodegenerative diseases”, however, table 1 didn’t include any neurodegenerative disease.

Response: Thank you for pointing this out. Table 1 provides major findings on ZBP1-mediated cell death and inflammation in various diseases. However, the detail mechanism of ZBP1 affecting neurodegenerative disease has not yet been identified. Thus, we have modified the description as “Besides, ZBP1 also plays an important role in regulating necroptosis in other human diseases, such as cancer and systemic inflammatory disease” in the revised manuscript (please see line 307). Thanks again for your kind help.